# Poisoning Attack on Federated Knowledge Graph Embedding

## ABSTRACT

Federated Knowledge Graph Embedding (FKGE) is an emerging collaborative learning technique for deriving expressive representations (i.e., embeddings) from client-maintained distributed knowledge graphs (KGs). However, poisoning attacks in FKGE, which lead to biased decisions by downstream applications, remain unexplored. This paper is the first work to systematise the risks of FKGE poisoning attacks, from which we develop a novel framework for poisoning attacks that force the victim client to predict specific false facts. The challenge is that FKGE maintains KGs for training locally on clients, preventing attackers in centralized KGEs from injecting poisoned data directly into the victim's training data. Thus, an attacker needs to create poisoned data without the victim's local KG, and inject the poisoned data indirectly into the victim's embeddings via FKGE aggregation. Specifically, to create poisoned data, the attacker first infers the targeted relations in the victim's local KG via a new KG component inference attack. Then, to accurately mislead the victim's embeddings via aggregation, the attacker locally trains a shadow model using the poisoned data and uses an optimised dynamic poisoning scheme to adjust the model and generate progressive poisoned updates. Our experimental results demonstrate the attack's effectiveness, achieving a remarkable success rate on various KGE models (e.g. 100% on TransE with WNRR), while keeping the original task's performance nearly unchanged.

## CCS CONCEPTS

• **Computing methodologies → Knowledge representation and reasoning**.

**ACM Reference Format:**
. 2024. Poisoning Attack on Federated Knowledge Graph Embedding . In *Proceedings of The Web Conference (WWW '24)*. ACM, New York, NY, USA, 10 pages. https://doi.org/XXXXXXX.XXXXXXX

## 1 INTRODUCTION

A Knowledge Graph (KG) is a structured knowledge repository that delineates real-world entities and their relationships through triples, where two entities act as nodes, and the relation between them serves as a directed edge. Numerous extensive KGs on the web, which are publicly accessible and collaboratively curated, including but not limited to Freebase [6], YAGO [32], and Wikidada [40], have been developed and employed in a wide range of downstream applications that harness the vast web-based knowledge. These KGs serve as invaluable resources for knowledge reasoning [19, 37, 51], recommendation systems [18, 46], and question-answering

**Unpublished working draft. Not for distribution.**

**Figure 1: An example for poisoning attack on FKGE. There are $m$ different clients, each of which uses its KG to train a local KGE model, and uses the model to output its entity embeddings $E^i$ ($i = 1 \dots m$) and relation embeddings $R^i$. In an FKGE training round, all clients send their entity embeddings to a server. The server aggregates all received embeddings and returns the result to all clients. The goal of the malicious server is to add a fake relation into the victim client's model.**

systems [2, 16, 22], enabling web applications to tap into a wealth of interconnected information.

Recent advances in representation learning techniques have accelerated the emergence of *KG embedding*, a process that maps KGs (i.e., entities and relations) into a unified embedding space, where each entity or relation is represented as a dense vector called an *embedding*. It can mitigate symbolic heterogeneity to facilitate diverse knowledge-driven applications [7, 38, 44]. This transformative approach has paved the way for developing powerful KG embeddings, enabling the representation of structured information in a continuous, high-dimensional vector space. An emerging research field, Federated KG Embedding (FKGE), takes KG embedding to the next level by harnessing Federated Learning (FL) principles alongside multi-source KGs to collaboratively enhance KG embedding [12, 19, 27, 52]. Based on FL, multiple KG owners can utilize the complementarity between different KGs to enhance their local models while preserving the sensitive KG data locally. This collaborative approach empowers organizations and researchers to collectively leverage the wealth of knowledge embedded in diverse KGs without compromising data privacy and security, thus opening up new frontiers for knowledge-driven applications and insights across domains.

However, the open collaboration among potentially self-interested parties in FL may pose new risks to FKGE. Some current studies have explored the privacy vulnerability of FKGE [19, 27, 50]. Their threat models tend to be *honest-but-curious*, i.e., they honestly follow the protocol but want to access others' sensitive data out of personal interest. Another type of attack that still remains unexplored is *poisoning attack*, which the latest FL systems focus on [9, 25, 30]. In particular, malicious participants can inject poisoned data or updates to the victim's model with the goal of reducing its model accuracy (i.e., untargeted poisoning attack) or implanting a backdoor in the model that can be exploited later, which forces the model to predict specific wrong facts (i.e., targeted poisoning attack). In

FKGE, we focus on targeted poisoning attack, which aims to add poisoned triples to the victim's model, leading to biased KGEs and incorrect decisions of the downstream applications.

**Example:** *In Figure 1, m hospitals as clients want to build a medical FKGE system, a pharmacist bribes a malicious server to manipulate the victim client (i.e., client 1) into predicting the outcome* $(Tom, is\ allergic\ to, aspirin)$*, which results in the doctor prescribing penicillin to Tom.*

In summary, this type of poisoning attack can be represented as adding a *fake relation* to the victim client's local KG and being learned by its model, which needs to address two challenges:

- **Unknown KG component.** To add a fake relation to the victim client's local KG, the attacker needs to know some of the components of the KG, including the targeted entities and the relations between them. However, this is difficult for the attacker because, in FKGE, only the entity embeddings are sent to the server.
- **Non-aggregatable relation embeddings.** To enable the local model of the victim client to learn the fake relation, the attacker needs to modify its relation embeddings maliciously. However, this is very difficult because, in FKGE, the server only aggregates entity embeddings from different clients and cannot manipulate any client relation embeddings.

Therefore, in this paper, we fill the gap in the absence of poisoning attacks in KFGE by designing a poisoning attack framework, which addresses the aforementioned challenges. The framework includes two attacks: server-initiate poisoning attack and client-initiate poisoning attack. To solve the first challenge, inspired by the privacy attack scheme in FKGE, in these two attacks, we design a new KG component inference attack to enable the malicious server or client to infer the original KG of the victim's client. Based on the known KG, the attacker can create a poisoned dataset with fake relations. To solve the second challenge, in these two attacks, we build a shadow KGE model on attacker, which is trained on the poisoned dataset and can indirectly affect the relation embeddings of the victim client by dynamically optimizing the shadow model and aggregating entity embeddings in the entire FKGE training process. Through these poisoning attacks, the malicious server or client can add fake relations into the victim client's local model without affecting the original task. We extensively evaluate the poisoning attack in FKGE for several KGE models on four real-world knowledge graph benchmark datasets.

Our contributions can be summarized as follows.

- We conduct the first holistic study for the poisoning attack on FKGE and propose two attack schemes from both client and server perspectives, which can successfully make the victim client's model learn fake relations without knowing the victim client's training data.
- We formulate the proposed attack, which indirectly misleads the victim's embeddings via FKGE aggregation, as a new KGE optimization problem and solve it by generating progressive poisoned updates.
- We evaluate our attack on four real-world KG datasets and four FKGE models to demonstrate that our proposed attack

can achieve high attack performance under different experimental settings, achieving an 100% attack success rate on WNRR and an average attack success rate of over 67%.
- We discuss potential countermeasures that shed light on improving the current practice of FKGE and point to several promising research directions, such as decentralized and verifiable KGE.

## 2 RELATED WORK

### 2.1 Federated Knowledge Graph Embedding

FKGE combines the principles of KGE with FL. It involves training embeddings for entities and relations from multiple distributed KGs while keeping them decentralized [11, 14, 19, 27, 52]. The first FKGE framework is FedE [12], which aggregates locally computed updates of entity embeddings to make the client learn from others' knowledge. Following FedE, some work has proposed other aggregation methods to improve the performance and robustness of FKGE [13, 27, 50, 52]. For example, FedLU [52] is an FL framework for heterogeneous KG embedding learning and unlearning that uses mutual knowledge distillation to transfer local knowledge to the global and absorb global knowledge back. Some current works pay attention to the privacy threats on FKGE. For example, Hu *et al.* [19] propose triple inference attacks on FKGE and design a differential privacy-based defence scheme to protect client's membership information. However, to the best of our knowledge, the vast majority of FKGE's work does not take into account the malicious settings and threat models of the participants, and there is no prior work exploring poisoning attacks in FKGE.

### 2.2 Poisoning Attack

Poisoning attacks involve the manipulation or injection of malicious data into a training dataset to compromise the performance and integrity of machine learning models. Existing works have achieved successful poisoning attacks against various scenarios, such as computer vision [15, 21] and natural language processing [26, 45]. In particular, in the context of open-source KG, some works have implemented poisoning attacks on centralized KGE models [3, 4, 48, 49]. For instance, MaSS [48] proposes a model-agnostic semantic and stealthy data poisoning attack on KGE models, which inserts indicative paths instead of triples to mislead the target KGE model, maintaining the effectiveness and stealthiness of poisoned datasets. However, these works against centralized KGE architecture by feeding poisoned data to the server responsible for training the model. Compared with the attacks on centralized KGE, data poisoning attacks against FKGE is more difficult because: (1) Because the raw KG data is stored locally on different clients, the attacker in FKGE is unable to know and change the training data of the victim, which makes it very difficult to build a poisoned dataset; (2) In FKGE, the server is only responsible for aggregating entity embeddings but not relation embeddings, thus the attacker in FKGE is unable to modify all embeddings of the victim, which makes it more difficult into inject poisoned data to the victim's model accurately.

## 3  PRELIMINARIES

**Knowledge Graph and Embedding.** A KG $\mathcal{G}$ includes many entities and their relationships. A triple $(h, r, t) \in \mathcal{T}$ is a fundamental unit of $\mathcal{G}$, where a head entity $h$ and a tail entity $t$ is connected by a relation $r$, and $\mathcal{T}$ is the triple set of $\mathcal{G}$. Knowledge graph embedding is a foundational technique in knowledge representation, aiming to project entities and relations from a KG $\mathcal{G}$ into continuous vector spaces. A KGE model learns the $d$-dimensional representations $X \in \mathbb{R}^d$ of the entities $\mathbf{e} \in X$ and the relations $\mathbf{r} \in X$. The general objective of KGE is to preserve the structured relational information of KG by a scoring function $g$, which represents the plausibility for each triple $(h, r, t)$. Some well-known models like TransE [7], DistMult [44], and ComplEx [35] are used as scoring functions in KGE. For example, in TransE, the scoring function $\mathcal{S}_{\text{TransE}}$ is defined as $g_\theta(h, r, t) = -\|\mathbf{h} + \mathbf{r} - \mathbf{t}\|$, where $\theta$ is the model parameters, $(\mathbf{h}, \mathbf{r}, \mathbf{t})$ are the embedding of $(h, r, t)$. The ultimate goal is to learn embeddings that minimize the score for real triples and maximize it for fake ones, allowing the model to make accurate predictions and infer missing information in the KG. Therefore, the loss function of KGE model can be represented as:

$\mathcal{L}(\mathbf{h}, \mathbf{r}, \mathbf{t}) = -\log \sigma \left( g_\theta(h, r, t) - \gamma \right) - \sum_{i=1}^{n} p_i \log \sigma \left( \gamma - g_\theta(h, r, t_i') \right),$

where $\gamma$ is the margin, $(h, r, t_i') \notin \mathcal{T}$ is a negative triple generated by replacing the original tail entity with a random entity, $n$ is the number of negative triples, and $p_i$ is the weight.

**Federated Knowledge Graph Embedding.** In FKGE, there exist a total of $m$ KGs, denoted as $\{\mathcal{G}_i\}_{i=1}^{m}$, where each KG may have overlapping entity sets and is privately held by an individual client. During the $\kappa$ round of FKGE training, each client, indexed as $i$, performs local updates on its respective KG $\mathcal{G}_i$, local relation embeddings $\mathbf{R}_{\kappa-1}^i$, and local entity embeddings $\mathbf{E}_{\kappa-1}^i$ over a certain number of iterations. Subsequently, the client transmits the updated local embeddings, denoted as $\mathbf{E}_\kappa^i$, to the central server. The server receives these updates from all clients, $\{\mathbf{E}_\kappa^i\}_{i=1}^{m}$ with $i$ ranging from 1 to $m$, and performs aggregation before broadcasting the resulting global embedding, $\mathbf{E}_\kappa$, back to all clients. These communication rounds are iteratively repeated until convergence is achieved.

## 4  OVERVIEW

In this section, we first introduce the system and threat model, then formulate the problem formulation of poisoning attacks in FKGE, and finally discuss the attack settings.

## 4.1  System and Threat Model

Like the vast majority of existing FKGE architectures [12, 19, 50], our system follows the most commonly used FedAvg algorithm [24] pattern in FL, adopting single server and multi clients settings. The tasks of the server and clients are introduced as follows:

- **Server.** A server's role includes the aggregation of entity embeddings collected from various clients and the subsequent transmission of these aggregated entity embeddings back to each respective client. Additionally, this server is tasked with the responsibility of upkeeping a comprehensive entity table. This table is utilized to log all distinct entities originating from various clients and to establish mappings between entities from different clients and the entries within this table.

- **Clients.** Different clients maintain unique knowledge graphs that contain overlapping entities, with each graph defining its own triples and relation sets. Through the updating operation, these separate clients utilize their corresponding triples to update their entity and relation embeddings.

Especially, we assume the following threat models:

*Server as Adversary:* As assumed in some FL systems [5, 29, 30, 36, 47], the server is not always trustworthy. It may forge or tamper with aggregation results and return poisoned embeddings for various reasons, such as program glitches, security vulnerabilities, and commercial interests. To ensure the availability of the system and the concealment of attacks, we assume that a malicious server can only send poisoned aggregation results to victim clients and correct aggregation results to other clients.

*Client as Adversary:* The malicious client has its local KGE model and dataset, it may add poisoned triples to its local dataset and transfer the poisoned aggregation results to other clients by uploading malicious embeddings to the server. It can collude with the server to some extent (even if the server is benign), e.g., asking the server which other clients have overlapping entities with it.

## 4.2  Problem Formulation

**Adversary's Objective.** In this study, we investigate the vulnerability of FKGE and design successful poisoning attacks that can mislead FKGE to add fake relations to victim client's local model. The goal of the attacker $\mathcal{A}$ is to minimize the score of the scoring function for triple $(h^*, r^*, t^*)$ as $\min g_{\hat{\theta}}(h^*, r^*, t^*)$, where $h^*, t^* \in \mathcal{E}$, $r^* \in \mathcal{R}$, $\mathcal{E}$ and $\mathcal{R}$ are the entity set and relation set of the victim client, and $(h^*, r^*, t^*) \notin \mathcal{T}$.

**Adversary's Knowledge.** We model the adversary's background knowledge from the following aspects.

- **Entity set and embeddings.** When a malicious server acts as an attacker, it has the entire entity set, each client's entity set and the periodically uploaded entity embedding matrices from all clients. When a malicious client acts as an attacker, it has the entire entity set, the entire entity embeddings (i.e., aggregated results), but does not have any other client's entity set and embeddings.

- **KGE models.** When a malicious server acts as an attacker, it knows the types of all client's KGE models and their partial model parameters, i.e., entity embeddings. When a malicious client acts as an attacker, it can only knows the types of all client's KGE models.

**Adversary's Capability.** When a malicious server acts as an attacker, we consider two capabilities:

- **Access to auxiliary data.** The adversary has access to an auxiliary KG dataset originating from the same domain as the FKGE learning process. In real-world scenarios, this auxiliary KG dataset can be sourced from publicly accessible repositories (e.g., Wikipedia) or constructed based on empirical common sense (e.g., establishing relations like patient and disease diagnosis).

- **Train a shadow model.** The adversary has the ability to train a shadow model using auxiliary datasets. This is not a special setting for the attack scheme, and the server can build a shadow dataset for the following purposes: 1) fine-tuning

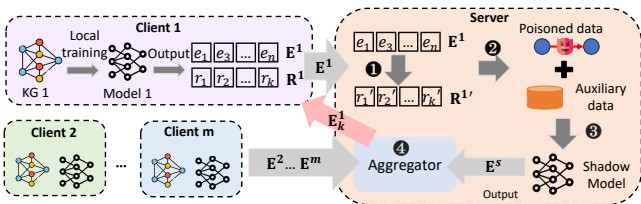

**Figure 2: Workflow of the server-initiate poisoning attack in FKGE.**

the global model by the shadow model, 2) serving as a source of regularization during aggregation, and 3) helping balance the learning process by providing additional information.

When a malicious client acts as an attacker, it can only use its own local KG and KGE model.

## 5 METHODOLOGY

In this section, we introduce two attacks in FKGE, including a server-initiate poisoning attack and a client-initiate poisoning attack.

### 5.1 Server-Initiate Poisoning Attack

In the server-initiate poisoning attack, the adversary first infers the local real relation set of the victim client and determines the existence of the relation between the targeted head and tail entities. Then, an auxiliary dataset and a shadow model are used to dynamically adjust the aggregation results to add fake relations to the local model of the victim client. The detailed attack process is shown in Figure 2, including the following four steps:

**Step1: Relation Inference.** In a certain FKGE training round, the victim client (e.g., client 1 in Figure 2) sends its entity embedding matrix to the malicious server. A previous work has proven that the server can infer the existence of real relations between these entities based on received entity embeddings [19]. The server first enumerates all potential relations between entities by calculating the scoring function of the KGE model. For example, in TransE, if a potential relation $r'$ is a plausible relation between a head entity $h'$ and a tail entity $t'$, its embedding will be close to $\|\mathbf{h}' - \mathbf{t}'\|$. The previous work has also noted that real relations tend to exhibit greater concentration within the embedding space, whereas fake relations typically display a more scattered distribution. Therefore, the malicious server can cluster potential sets of relations into some clusters, and identifies the relation embeddings near the concentrated cluster centers as real relations. Furthermore, the malicious server can use its auxiliary dataset to infer the original relations corresponding to these real relation embeddings.

**Step2: Poison Data Generation.** After inferring the real relations of the victim client, the malicious server first determines whether there is a relation between the targeted head entity $h^*$ and tail entity $t^*$. If it does not exist, the server chooses a relation $t^*$ from $\mathcal{R}$, where $\mathcal{R}$ is the the victim client's relation set that the server has inferred in **Step1**. The server then adds the poisoned triple $(h^*, r^*, t^*)$ into the auxiliary dataset $\mathcal{D}_a$. We define the auxiliary dataset with the poisoned triples as $\mathcal{D}'_a$. To make the poisoned dataset more pure (i.e., reducing the impact of unrelated triples on the FKGE model), the server removes data unrelated to the victim client's local dataset from the auxiliary dataset, leaving only the

victim client's raw data and poison triples in the auxiliary dataset. The server uses the purified poisoned dataset as the training dataset $\mathcal{D}_p$ to train a shadow model, and the training dataset $\mathcal{D}_p$ can be represented as $\mathcal{D}_p = \{\mathcal{T}_1 \cap t_p\}$, where $\mathcal{T}_1$ is the triple set of client 1 and $t_p = (h^*, r^*, t^*)$.

**Step3: Shadow Model Training.** To imitate the victim client's KGE model for learning poisoning data, the malicious server builds a shadow model $f_\theta()$ that is a KGE model trained from the poisoning dataset $\mathcal{D}_p$ and of the same type as the client's model. The malicious server optimizes the following function to training the shadow model $f_\theta()$:

$$\min f_{\hat{\theta}}\left(h^*, r^*, t^*\right),$$
$$\text{s.t., } \hat{\theta} = \arg\min_\theta \sum_{(h,r,t) \in \mathcal{T}_1} f_\theta(h, r, t). \tag{1}$$

After the training process, the malicious server obtains a well-designed shadow model that can give a large plausibility for the poisoned triple.

**Step4: Embedding Aggregation.** In the aggregation process, the malicious server first aggregates the entity embeddings of all clients and obtains the aggregate result $\mathrm{E}_\kappa$. Then, to add the poisoned data into the victim client's local model, the server uses its shadow model to adjust the aggregation result. An intuitive method is to directly aggregate $\mathrm{E}_\kappa$ and the entity embeddings output by the shadow model $\mathrm{E}^s_\kappa$. We name this method *fixed model poisoning*. However, this strategy may not achieve a good attack success rate because the aggregation result only affects the entity embeddings of the victim client and does not mislead its relation embeddings, which results in the victim client model having lower confidence in predicting poisoned triples than the shadow model.

To address the issue, we design another attack method called *dynamic poisoning*, which indirectly misleads the relation embeddings of the victim client by dynamically optimizing the shadow model during FKGE's training process. In dynamic poisoning attack, the goal of the server is not only to send poisoned entity embeddings to the victim client, but also to indirectly enable the victim client to learn poisoned relation embeddings through the aggregation results. Recall that in **Step1**, the malicious server has infered the relation embeddings of the victim client. Therefore, the server can dynamically optimize the shadow model during each round of FKGE training by calculating the victim client model's scoring of the poisoned triple. The overall optimization objectives are as follows:

$$\arg\min_{\hat{\theta}, \hat{\theta}'} \mathcal{L}_{\hat{\theta}, \hat{\theta}'}\left(t_p\right),$$
$$\mathcal{L}_{\hat{\theta}, \hat{\theta}'} = f_{\hat{\theta}}\left(t_p\right) + g_{\hat{\theta}'}\left(t_p\right) + \mathcal{L}\left(f_{\hat{\theta}}\left(t_p\right), g_{\hat{\theta}'}\left(t_p\right)\right).$$
$$\text{s.t., } \hat{\theta} = \arg\min_\theta \sum_{(h,r,t) \in \mathcal{T}_1} f_\theta(h, r, t), \tag{2}$$
$$\hat{\theta}' = \arg\min_{\theta'} \sum_{(h,r,t) \in \mathcal{T}_1} g_{\theta'}(h, r, t),$$

where $t_p$ is the poisoned triple $(h^*, r^*, t^*)$.

**Overall Training.** The algorithm 1 presents the overall training process of the server-initiate poisoning attack in FKGE, where $\mathcal{L}_{\hat{\theta}, \hat{\theta}'}$ is described in Equation 2. In the FKGE training process, the server

---

**Algorithm 1:** Server-initiate Poisoning Attack in FKGE

**Input** : Victim client's $c_v$ and its clean model $g_{\theta'}$, m clients with m KGs $\{\mathcal{G}_i\}_{i=1}^m$, a shadow model $f_\theta$, an auxiliary dataset $\mathcal{D}_a$, communication rounds $T$

**Output**: Victim client's poisoned model $g_{\hat{\theta}'}$

1   Server initializes $\mathbf{E}_0$.

2   **for** *round = 1, . . . , T* **do**

3     Each client sends its local entity embeddings $\mathbf{E}_{round}^i$ of its KG $\mathcal{G}_i$ to the server.

4     In the first round of the attack, the server infers $c_v$'s relation embeddings $\mathbf{R}^v$, create poisoned triple $t_p = (h^*, r^*, t^*)$, and construct its train dataset $\mathcal{D}_p = \{\mathcal{T}_v \cap t_p\}$.

5     The server uses $\mathcal{D}_p$ to train and $\mathbf{R}^v$ to dynamically optimize $f_{\hat{\theta}}$.

6     $\hat{\theta} = \arg\min_{\hat{\theta},\hat{\theta}'} \mathcal{L}_{\hat{\theta},\hat{\theta}'}(t_p)$.

7     $\mathbf{E}_{round} = \text{aggregate}(\mathbf{E}_{round}^1, \ldots, \mathbf{E}_{round}^m)$.

8     $\mathbf{E}_{round'} = \text{aggregate}(\mathbf{E}_{round}, \mathbf{E}_{round}^s)$.

9     The server returns $\mathbf{E}_{round'}$ to $c_v$ and $\mathbf{E}_{round}$ to other clients.

10    The victim client update its model $g_{\hat{\theta}'}$.

11 **return** $g_{\hat{\theta}'}$

---

first initializes a global entity embeddings matrix randomly and sends it to all clients (line 1). In each round, all clients send their entity embeddings to the server (line 3) and the server can initiate the inference attack in any round (line 4). The difference between the fixed model poisoning attack and the dynamic poisoning attack is reflected in lines 5 and 6, where the dynamic poisoning attack needs to dynamically optimize the shadow model. Finally, the server returns the aggregation results to clients and clients update their models (lines 7-10).

### 5.2 Client-Initiate Poisoning Attack

In the client-initiate poisoning attack, the malicious client first infers the local real relation set of the victim client and determines the existence of the relation between the targeted head and tail entities. Then, the malicious client uses its local KG and KGE model to add fake relations to the local model of the victim client. The difference between the server-initiate poisoning attack and the client-initiate poisoning attack is that the malicious client cannot obtain the entity set of the victim client.

Therefore, the malicious client follows the four steps shown in subsection 5.1 to launch a poisoning attack, but with the following differences: 1) in **step1**, the malicious client needs to ask the server about the overlap between it and the victim client entity set. It needs to infer whether there is a relation between the targeted head and tail entities in the victim client's dataset based on the changes in its local relation embeddings during the training process of FKGE, which has been proven feasible in the previous work [19]; 2) in **step2**, **step3** and **step4**, the malicious client uses its local KGE model to replace the shadow model, and uses its local relation embeddings to simulate the relation embeddings of the victim client to dynamically optimize its local model.

### 5.3 Potential Defense Mechanism

*5.3.1 Server-initiate Poisoning Attack Defense.* Due to the data isolation of FKGE, i.e., it is difficult for victim client to distinguish whether the poisoned aggregation results come from malicious servers or other benign clients, the proposed attack cannot be detected by existing error detection methods. By analyzing the workflow of the attack, we find that the most effective defense method is to prevent the inference attack from the malicious server. As long as the malicious server is unable to obtain the relation embeddings of the victim client, its attack will fail. Some previous work has attempted to use differential privacy to defend against inference attacks, i.e., adding controlled noise to the data or model parameters to prevent malicious servers from aligning the raw data for analysis. For example, DPSGD[1] and DP-FLames [19] have been proven to be effective in defending against inference attacks in FKGE.

*5.3.2 Client-initiate Poisoning Attack Defense.* Similar to the server-initiate poisoning attack, it is difficult for victim client to distinguish whether the poisoned aggregation results come from malicious clients or other benign clients. In addition to the differential privacy-based defense mechanism, we explore another new paradigm for FKGE, i.e., the decentralized knowledge graph embedding (DKGE), by using blockchain instead of the centralized server to make the entire training process of KGE verifiable. In any training round of DKGE, each client uploads its entity embedding updates to the blockchain in the form of blockchain transactions, such as smart contract transactions in Ethereum [41]. Then, to aggregate the embedding updates, each client downloads embeddings that overlap with some of their own entities on the blockchain and aggregates them. To accelerate aggregation efficiency, we adopt *asynchronous aggregation* in DKGE, which means that any client can upload and download embedding updates at any time. Due to the independent operation of the aggregation process by the client and the immutability of the blockchain, malicious participants are easily detected by victims. Furthermore, to protect the privacy of clients and further reduce their space for wrongdoing, we suggest that developers of the DKGE system use zero-knowledge proof [8, 39, 42] (ZKP) technology to allow clients to prove their local data and operations without compromising privacy, or use privacy set intersection (PSI) [10, 20, 28] to perform overlapping entity calculations without compromising privacy. We implemente a simple DKGE prototype and make its more concrete implementation our future work.

## 6 EVALUATION

In this section, we test the effectiveness of our proposed attacks on four benchmark datasets in federated settings, targeting four state-of-the-art KGE models. Specifically, our evaluations aim to address the following research questions:

**RQ1** Can our poisoning attacks effectively enhance the predictions of the KGE model for the targeted victim client on the poisoned triples?

**RQ2** To what extent will the original link prediction performance of the targeted client be affected following the execution of a poisoning attack?

**Table 1: Statistics of Four Datasets.**

|          | FB15k237 | NELL995 | WN18RR | CoDEx-M |
|----------|----------|---------|--------|---------|
| Entities | 14951    | 75492   | 40493  | 17050   |
| Relations| 237      | 200     | 11     | 51      |
| Triples  | 272115   | 149678  | 86835  | 185584  |

**RQ3** How do different settings affect the effectiveness of the attack, including the number of poisoned triples and the number of clients in FKGE?

**RQ4** Can potential defense mechanisms mitigate the effectiveness of the attack?

In a word, we evaluate the efficacy of the attack strategy in enhancing the KGE model's predictions of the victim client on the poisoned triples while simultaneously preserving the original performance of all other benign clients as much as possible.

## 6.1 Experiment Setups

**Datasets.** To evaluate the effectiveness of our attack, we utilize four publicly available benchmark knowledge graph datasets - *FB15k237* [34], *NELL995* [43], *WN18RR* [7], and *CoDEx-M* [31]. In order to conduct our evaluation in a federated setting, we create client datasets as described in [12]. Specifically, we randomly select relations for each client and distribute triples into the clients based on the chosen relations. We randomly split dataset into 2, 3, 4, 5 clients as dataset-Fed2,-Fed3,-Fed4,-Fed5.

In order to perform poisoning attacks, the attacker needs to create poisoned datasets to train the shadow model. Initially, the server randomly selects $m$ head entities as the poisoned head entities for the victim client. Subsequently, the server randomly chooses a tail entity that has no relation with the selected head entity. Finally, the server inserts a fake relation between the chosen head entity and the selected tail entity to form the poisoned triples. The poisoned dataset consists of an auxiliary dataset and the poisoned triples. The detailed statistics of the original datasets are given in Table 1.

**Victim Models.** We select four state-of-the-art KGE models, namely TransE [7], RotatE [33], ComplEx [33] and DistMult [33], as the victim models. The attacker adopts the k-means clustering [23] for the inference attack. As for the shadow model, the server employs the same type of KGE model as the clients. As introduced in subsection 5.1, the shadow model is trained on the poisoned datasets. We train FedE [12] on the orginal dataset as baseline to compare the performance of our attack. For the implementation of our attacks, we follow FedE to set hyperparmeters. The local training epoch for the client model is set to 3 and we evaluate the attack performance using the validation set every 5 rounds. We adopt early termination, which means if the model's MRR performance on the validation set remains unchanged after 5 rounds, we terminate the training process and save the best model parameters.

**Evaluation Metrics.** We report Mean Reciprocal Rank (MRR) and Hits at $N$ (Hits@$N$, $N$ = 1, 5, 10) to validate the performance of link prediction on each client, which follows the common practice in KGE literature. Higher Hits@$N$ and MRR mean better KGE model prediction performance. To be able to further evaluate the effectiveness of attacks in increasing the prediction performance of the poisoned triples on victim client, we test the values of MRR and Hits@$N$ about the poisoned triples in clean settings and attack settings, where clean settings represent the prediction performance of the clean model.

## 6.2 Attack Evaluation

In this section, we demonstrate the effectiveness of the proposed poisoning attacks. First, we test the predictions of the KGE model for the targeted victim client on the poisoned triples (**RQ1**) and on the original link prediction task (**RQ2**). Second, we test the effectiveness of the attacks under different settings (**RQ3**).

### 6.2.1 Attack Performance (RQ1).

**Attack performance of malicious server**. We implement two malicious server attacks introduced in subsection 5.1, including the fixed model poisoning (FMPA-S) and dynamic poisoning (DPA-S). We randomly select a victim client for the fixed model poisoning attack. To ensure fair comparison, we consistently choose the same victim client when executing the dynamic poisoning attack. We select 10 head entities as the head entity of poisoned triples on the victim client and set the client number is 3. The MRR and Hit@$N$ values on the original task and poisoned triples are reported. The results are presented in Table 2.

In Table 2, the poisoned triples on victim model (PT on VC) column clearly illustrates that our proposed attack methods, FMPA-S and DPA-S, significantly enhance the link prediction performance on poisoned triples compared to the original FedE. It is worth noting that in most cases, DPA-S outperforms FMPA-S in terms of link prediction accuracy on poisoned triples. When utilizing TransE, RotatE, ComplEx, and DistMult as KGE models, DPA-S achieves an average MRR of 0.67 and Hits@10 of 0.88 on the poisoned triples in dataset-Fed3 (FB15k237-Fed3, NELL995-Fed3, WN18RR-Fed3, and CoDEx-M-Fed3). Conversely, FMPA-S achieves an average MRR of 0.56 and Hits@10 of 0.79 on the poisoned triples in dataset-Fed3. Furthermore, all the MRR and Hit@$N$ values for poisoned triples under FedE settings are found to be 0. For example, when using TransE as the KGE model, the Hit@10 exceeds 0.9 on the FB15k-237 dataset under the DPA-S attack, indicating that over 90% of the poisoned triples on the victim client are predicted within the top-10 of the ranking list. Additionally, we can conclude that the WNRR dataset is more vulnerable to poisoning attacks compared to other datasets due to its sparser structure,i.e., it has fewer neighbors per triple. By injecting few triples, the poisoning attacks can achieve a high attack success rate.

**Attack performance of malicious client**. We randomly select a malicious client and a victim client to evaluate the attack performance of the malicious client. We set the number of poisoned triples to be 10, and the link prediction results of the poisoned triples on the victim clients are presented in Table 3. Specifically, the malicious client trains its local KGE model using the poisoned dataset. The poisoned dataset consists of the original dataset of the malicious client and the poisoned triples inferred based on the changes in the local relation embeddings of the malicious client. As shown in Table 3, the client poisoning attack (CPA) achieves an average MRR of 0.59, Hits@10 of 0.81 on the poisoned triples in dataset-Fed3. In addition, we also observe that the client poisoning attack is more effective on the WNRR dataset compared to other datasets, similar to what is observed in the poisoning attack on the

**Table 2: Attack Performance of Server-initiate Poisoning Attack (PT on VC means poisoned triples on victim model).**

| Dataset | Model | TransE Mean MRR | TransE Mean Hit@10 | TransE PT On VC MRR | TransE PT On VC Hit@10 | RotatE Mean MRR | RotatE Mean Hit@10 | RotatE PT On VC MRR | RotatE PT On VC Hit@10 | DsitMult Mean MRR | DsitMult Mean Hit@10 | DsitMult PT On VC MRR | DsitMult PT On VC Hit@10 | ComplEx Mean MRR | ComplEx Mean Hit@10 | ComplEx PT On VC MRR | ComplEx PT On VC Hit@10 |
|---|---|---|---|---|---|---|---|---|---|---|---|---|---|---|---|---|---|
| FB15k-237 | FedE | 0.41 | 0.60 | 0.00 | 0.00 | 0.41 | 0.60 | 0.00 | 0.00 | 0.38 | 0.55 | 0.00 | 0.00 | 0.38 | 0.55 | 0.00 | 0.00 |
| | FMPA-S | 0.40 | 0.59 | 0.46 | 0.80 | 0.41 | 0.60 | 0.42 | 0.80 | 0.38 | 0.55 | 0.14 | 0.40 | 0.38 | 0.55 | 0.33 | 0.70 |
| | DPA-S | 0.40 | 0.59 | **0.56** | **0.90** | 0.41 | 0.60 | **0.43** | **0.80** | 0.38 | 0.55 | **0.25** | **0.50** | 0.38 | 0.55 | **0.39** | **0.80** |
| NELL995 | FedE | 0.71 | 0.87 | 0.00 | 0.00 | 0.75 | 0.88 | 0.00 | 0.00 | 0.28 | 0.45 | 0.00 | 0.00 | 0.37 | 0.54 | 0.00 | 0.00 |
| | FMPA-S | 0.69 | 0.87 | 0.45 | 0.60 | 0.74 | 0.87 | 0.70 | 0.80 | 0.26 | 0.42 | 0.60 | 0.70 | 0.35 | 0.52 | **0.66** | 0.80 |
| | DPA-S | 0.69 | 0.86 | **0.52** | **0.80** | 0.74 | 0.85 | **0.83** | **0.90** | 0.26 | 0.42 | **0.65** | **0.70** | 0.33 | 0.51 | 0.63 | **0.90** |
| WNRR | FedE | 0.18 | 0.37 | 0.00 | 0.00 | 0.25 | 0.39 | 0.00 | 0.00 | 0.17 | 0.21 | 0.00 | 0.00 | 0.16 | 0.20 | 0.00 | 0.00 |
| | FMPA-S | 0.16 | 0.36 | 0.80 | 0.80 | 0.25 | 0.38 | 0.93 | 1.00 | 0.15 | 0.18 | 0.77 | 0.90 | 0.17 | 0.22 | 0.95 | 1.00 |
| | DPA-S | 0.16 | 0.34 | **1.00** | **1.00** | 0.24 | 0.37 | **1.00** | **1.00** | 0.15 | 0.19 | **1.00** | **1.00** | 0.15 | 0.18 | **1.00** | **1.00** |
| CoDEx-M | FedE | 0.52 | 0.75 | 0.00 | 0.00 | 0.53 | 0.77 | 0.00 | 0.00 | 0.46 | 0.67 | 0.00 | 0.00 | 0.46 | 0.67 | 0.00 | 0.00 |
| | FMAP-S | 0.50 | 0.73 | 0.30 | 0.9 | 0.52 | 0.75 | 0.36 | 0.90 | 0.44 | 0.66 | 0.60 | 0.80 | 0.46 | 0.68 | 0.50 | 0.80 |
| | DPA-S | 0.50 | 0.73 | **0.61** | **1.00** | 0.52 | 0.75 | **0.58** | **1.00** | 0.44 | 0.65 | **0.67** | **0.90** | 0.46 | 0.68 | **0.65** | **0.90** |

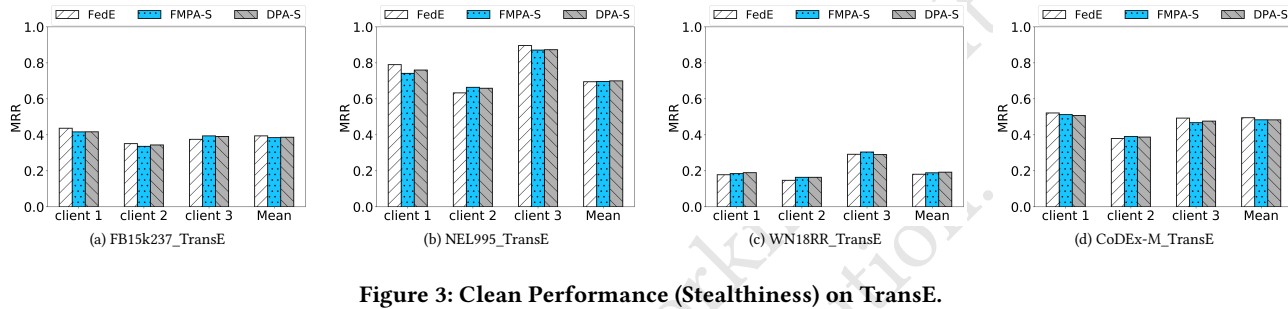

Figure 3: Clean Performance (Stealthiness) on TransE.

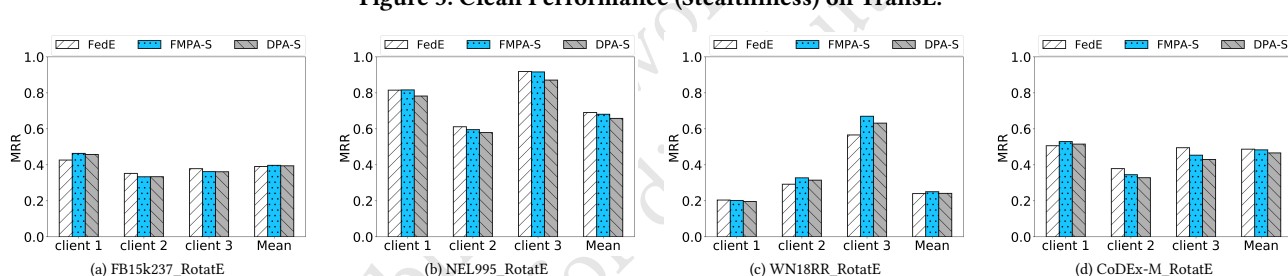

Figure 4: Clean Performance (Stealthiness) on RotatE.

server side. CPA achieves an average MRR of 0.64 and Hits@10 of 0.88 on the poisoned triples in WNRR-Fed3.

In summary, these results demonstrate that our attack methods can efficiently elevate the ranks of poisoned triples, posing severe threats to knowledge graph embedding.

*6.2.2 Clean Performance (Stealthiness) (**RQ2**).* We investigate the original test link prediction performance of different clients under our poisoning attack to validate the stealthiness of the attack. Our goal is to test how much the original link prediction performance remains unchanged. The MRR values of four dataset on TransE and RotatE are shown in Figure 3 and Figure 4. In the results, the performance differences between DPA-S, FMPA-S, and FedE are small across all local clients. This demonstrates that our attack can effectively balance the performance of the attack while maintaining the original link prediction performance as closely as attainable.

*6.2.3 Comparison of Different Settings (**RQ3**).*
**Impact of the number of clients**. We explore the attack performance with different numbers of clients. Specifically, using TransE

as the KGE model, we test the MRR and Hit@$N$ values on the CoDEx-M dataset, varying the number of clients from 2 to 5. As shown in Figure 5, the metric values generally decrease as the number of clients increases. We speculate that the decreased attack effectiveness stems from the fact that the server's aggregated operations become diluted as the number of clients increases.

**Impact of the number of poisoned triples.** We use different poisoned datasets to investigate whether the effectiveness of the poisoning attack increases with the number of poisoned triples. The MRR and Hit@$N$ results on CoDEx-M-Fed3 using the RotatE model as the KGE model are depicted in Figure 6. We vary the number of poisoned triples from 0 to 150. From the Figure 6, the metric values of on poisoned triples fluctuates within a certain range in CoDEx-M dataset. Therefore, under our settings, the number of poisoned triples does not have a significant impact on the attack success rate.

## 6.3 Defense Evaluation (RQ4)

We finally test the effectiveness of the defense mechanisms introduced in subsection 5.3. For the server-initiate poisoning attack

**Table 3: Attack Performance of Client-initiate Poisoning Attack (PT on VC means poisoned triples on victim model).**

| Dataset | TransE | | | | RotatE | | | | ComplEx | | | | DistMult | | | |
|---|---|---|---|---|---|---|---|---|---|---|---|---|---|---|---|---|
| | Mean | | PT On VC | | Mean | | PT On VC | | Mean | | PT On VC | | Mean | | PT On VC | |
| | MRR | Hit@10 | MRR | Hit@10 | MRR | Hit@10 | MRR | Hit@10 | MRR | Hit@10 | MRR | Hit@10 | MRR | Hit@10 | MRR | Hit@10 |
| FB15K-237 | 0.41 | 0.60 | 0.68 | 0.90 | 0.41 | 0.61 | 0.58 | 0.80 | 0.37 | 0.55 | 0.36 | 0.70 | 0.38 | 0.56 | 0.41 | 0.60 |
| NELL995 | 0.68 | 0.85 | 0.85 | 0.90 | 0.76 | 0.88 | 0.77 | 0.90 | 0.47 | 0.61 | 0.47 | 0.70 | 0.32 | 0.48 | 0.38 | 0.50 |
| WNRR | 0.17 | 0.38 | 0.66 | 0.80 | 0.27 | 0.39 | 0.53 | 0.90 | 0.15 | 0.20 | 0.61 | 0.90 | 0.16 | 0.20 | 0.74 | 0.90 |
| CoDEx-M | 0.51 | 0.75 | 0.52 | 0.90 | 0.52 | 0.77 | 0.71 | 0.80 | 0.46 | 0.67 | 0.74 | 0.80 | 0.45 | 0.67 | 0.39 | 0.90 |

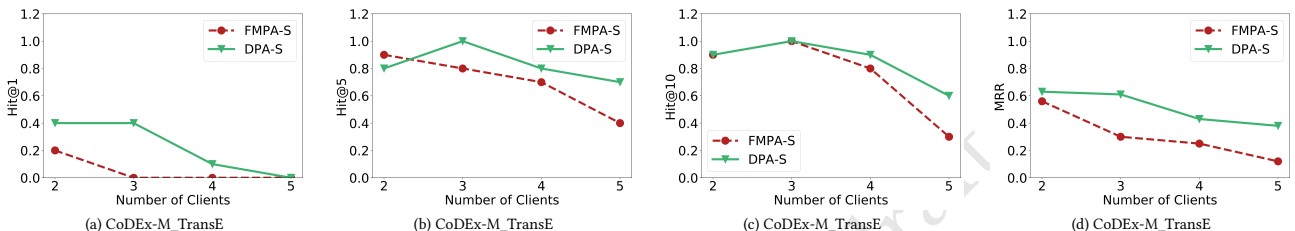

**Figure 5: Attack Performance of Different Numbers of Clients.**

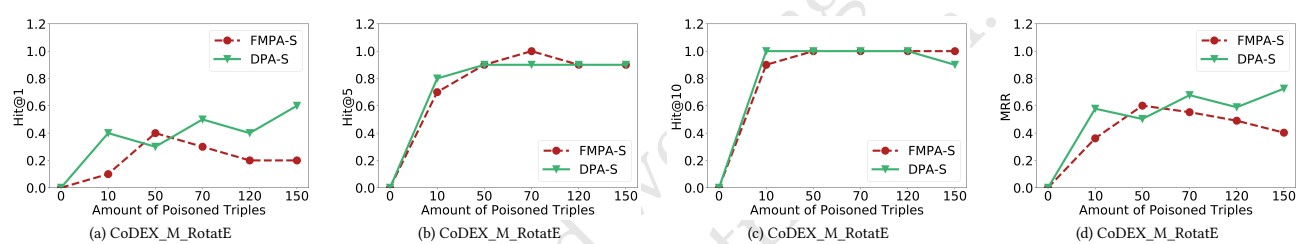

**Figure 6: Attack Performance of Different Numbers of Poisoned Triples.**

**Table 4: Attack Performance after Defense**

| Dataset | Schemes | MRR | Hit@10 |
|---|---|---|---|
| FB15k-237 | DPA-S | 0.56 | 0.90 |
| | DPSGD | 0.34 | 0.60 |
| | DP-Flames | 0.33 | 0.50 |
| CoDEx-M | DPA-S | 0.61 | 1.00 |
| | DPSGD | 0.31 | 0.60 |
| | DP-Flames | 0.30 | 0.50 |

defense, we adopt two differential privacy-based methods, DPSGD and DP-Flames, to defend against attacker's membership inference attacks and weaken its poisoning attacks. We test the MRR and Hit@10 values of three schemes on the model TransE and datasets FB15k-237-Fed3 and CoDEx-M-Fed3. The results are shown in Table 4. From Table 4, we can see that after adopting defense mechanisms to the FB15k-237-Fed3 dataset, the MRR value decreases from 0.56 to 0.34 and 0.33, and the Hit@10 value decreases from 0.90 to 0.60 and 0.50. On the CoDex-M-Fed3, the defense effectiveness is better. These results demonstrate that the differential privacy-based defense mechanisms can achieve certain defensive effects, but there is still significant room for research.

For the client-initiate poisoning attack defense, we implement a prototype of DKGE by replacing the server with an Ethereum blockchain. 5 clients is installed with go-ethereum [17] nodes and trained through asynchronous aggregation for FKGE. Although the attack behavior in the system can be correctly detected and traced back to the attacker's identity, the convergence speed of the system is significantly slower than the previous FKGE. Therefore, further designs are needed to improve its availability.

## 7 CONCLUSION

In this paper, we have comprehensively investigated the poisoning attacks of FKGE. Our attacks can accurately inject fake relations into the victim's model, even if their local KG data and some model parameters are unknown. We demonstrate the effectiveness and practicality of four real-world datasets and four KGE models. The dynamic poisoning attack achieves an average MRR of 0.67 and Hits@10 of 0.88 on the poisoned triples in four datasets on four different KGE models. In particular, the success rate of the attack is 100% on the WNRR dataset. Furthermore, the experimental results also demonstrate that the FKGE's original task performance is not significantly affected by our attacks. As for future studies, although the potential defence mechanisms can mitigate the attack effectiveness to some extent, they still have significant limitations and affect the system's availability. We will consider these weaknesses and plan to investigate a more secure FKGE architecture.

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
