# OpenReview forum: "Poisoning Attack on Federated Knowledge Graph Embedding"
_ACM.org/TheWebConf/2024/Conference — TheWebConf24 Oral_

### Official Review · Reviewer_GEha · 2023-11-22

**Novelty:** 4
**Technical Quality:** 5

**Review:**

This paper introduces a poisoning attack for knowledge graph embeddings that are shared in a federated environment. Essentially, it shows that in an environment where embeddings are shared between network nodes via a server both corrupted clients and servers can improve the performance on link prediction on corrupted relations on other clients. Evaluation is done using standard link prediction benchmarks with modifications to test the specific attack.

_Strengths_

1. An interesting idea for the attack especially the idea of using auxiliary data.
2. Clear description of the problem setting and the related work.
3. Interesting battery of experiments

_Weaknesses_

1. The case for the importance of this kind of attack is a bit unclear. Is federated training of embeddings where corrupting link prediction performance that common?
2. In places, it's not clear whether the malicious client and server are working together or if this is tested or not.

__After Rebuttal__
The comments addressed my questions in particular the additional experiments were helpful for understanding.

**Questions:**

- In Table 2 can you clarify how many clients are used?
- There's a suggestion that the increase in the number of clients decreases the effectiveness of the attack. Do you have a deeper rationale why that is? Can that mitigation be done without a distributed environment or does one have to have a distributed environment.
- Can you clarify if you test the case where a malicious client and server are colluding.
- Can yo provide a deeper justification of why this attack might occur and in particular what use cases?

**Reviewer Confidence:**

3: The reviewer is confident but not certain that the evaluation is correct

**Scope:**

3: The work is somewhat relevant to the Web and to the track, and is of narrow interest to a sub-community

---

### Official Review · Reviewer_H4Wx · 2023-11-23

**Novelty:** 5
**Technical Quality:** 6

**Review:**

Quality: the quality of the descriptions and experiments is mostly good, with some comments on the clarity below. Research questions are clearly formulated and give a good picture of the approach.

Clarity: mostly readability and clarity is good, but with a number of typos that could be easily avoided. I recommend fixing this to in crease the overall quality of the text

Originality: the approach is original and interesting - of course the indivual approaches have been used before in other context, but the result is in my opinion quite original

Signifiance: the evaluation give a clear picture of the significance, in all of the four research questions posed

Pros:

* Interesting, current topic

* Clear research questions and answers

Cons:

* Some care for the clarity (typos, ...) would improve overall appreciation

**Questions:**

I have in most parts a clear picture of the paper, and only minor questions:

1. You talk about "holistic" study - what in your opinion constitutes such a study (and what would not)?

2. In terms of the designed methodology, I understand very clearly the steps proposed. Could you give some additional comments on why you exactly chose this methodology compared to nearby alternatives?

**Ethics Review Description:**

/

**Reviewer Confidence:**

3: The reviewer is confident but not certain that the evaluation is correct

**Scope:**

4: The work is relevant to the Web and to the track, and is of broad interest to the community

---

### Official Review · Reviewer_6vj7 · 2023-11-23

**Novelty:** 6
**Technical Quality:** 6

**Review:**

The paper describes a poisoning attack on Federated Knowledge Graph Embedding as well as a defence against the poisoning attack. It considers a federation composed of participants with their own private KG and a central server. The n participants compute locally their graph embedding and send entity embeddings to a central server. The central server aggregates entity embeddings and redistributes them to participants. Shared entity embeddings help to have more accurate predictive models for each participant while keeping KG private.

The paper demonstrates how a malicious participant or malicious central server can add a fake relation in a participant model. For example, a malicious participant adds a relation (tom, is-allergic-to, aspiring) into another KG model, biassing all downstream applications on the victim model.

The attack is challenging as the malicious server or participant doesn't know the relations embeddings of the victim, and cannot modify directly relations embeddings of the victim. [19] already tackle the problem of discovering relation embedding, the originality of the paper is to incite the victim model to predict a fake relationship between 2 entities without modifying any relation embedding.

The paper describes precisely the attack and a possible defence. It evaluates the attack and the defence on a simulated setup. The experiment is conducted on several well-known datasets with different  graph embedding techniques.


Strong points:
* I consider that the problem is timely. Federated Learning with knowledge graphs is a very appealing context. Describing precisely how such an approach can be poisoned is very valuable for the community.

* The paper is very well-written, with strong motivations and convincing examples.

* Positioning vs State of the Art is clear : poisoning attacks exist on centralised Knowledge Graph Embeddings, not on federated knowledge Graph Embeddings. On Federated Knowledge Graph Embedding,  [19] described how it is possible to infer some relations from entity embeddings, but not to poison a participant model.

* The experimentations of attacks and defence are convincing

Weak points:
* It seems that the heart of the proposal is how to force a target model to predict a fake relation without modifying directly relation embeddings. This is done by training a shadow model on the central server or on a participant. Such a part is mainly explained by giving the function to train the shadow model (L419-455). I would know how, by just changing target entity embeddings, the target model is forced to predict a fake relation ? Can you elaborate more on this  ?

* From the experiments, It seems that the Defense is more complex than the attack, especially for the server-initiate attack. Is it possible for a participant just to detect that the server is attacking, maybe using the same blockchain as for the client-initiate attack (Q2) ??
I did not see any reference to public implementation of the attack and the defence prototypes (maybe I missed it ?). Are  the attack and defence  experiments reproducible thanks to a public repository (Q3) ??

Overall, I found the paper very interesting to read with a significant scientific contribution.

**Questions:**

*  I would know how, by just changing target entity embeddings, the target model is forced to predict a fake relation ? Can you elaborate more on this  ?
* Is it possible for a participant just to detect that the server is attacking, maybe using the same blockchain as for the client-initiate attack ?
* Are  the attack and defence  experiments reproducible thanks to a public repository ??

**Reviewer Confidence:**

3: The reviewer is confident but not certain that the evaluation is correct

**Scope:**

4: The work is relevant to the Web and to the track, and is of broad interest to the community

---

### Official Review · Reviewer_u75f · 2023-11-24

**Novelty:** 5
**Technical Quality:** 5

**Review:**

**Summary**:

This paper proposes a poison attack framework for Federated Knowledge Graph Embedding. The framework can lead to sever-initiate and client-initiate attacks. The authors also study the corresponding defense mechanism. The experimental results verify the superiority of the proposed attacks.

**Strengths**:
S1. The presentation of the paper is clear.

S2. The proposed attack framework is effective regarding the poisoned triples on victim model.

S3. The analysis of the framework is comprehensive, and two attack mechanisms and a defense method are well discussed.

**Weaknesses**:

W1. Based on Figure 5 and Figure 6, the attack performance does not steadily increase as the budget grows, reflecting the instability of the framework.

W2. Based on Figure 6, the attack seems to be effective when injecting a certain amount of poisoned triplets on some datasets, which could lead to unnoticability issues (which are not well discussed in the paper).

W3. The attack settings might be so impractical that the adversary requires the relation embeddings.

**Questions:**

1. What are the computational costs of the framework?

2. How do we ensure the auxiliary dataset originates from the same domain?

**Reviewer Confidence:**

2: The reviewer is willing to defend the evaluation, but it is likely that the reviewer did not understand parts of the paper

**Scope:**

3: The work is somewhat relevant to the Web and to the track, and is of narrow interest to a sub-community

---

### Decision · Program_Chairs · 2024-01-22

**Decision:**

Accept (Oral)

**Comment:**

This article introduces a poison attack framework for Federated Knowledge Graph Embedding, which can trigger server and client attacks.
 Using this framework, various defense techniques were investigated.

 All reviewers agree that this is a valuable contribution for the Web Conference, and deserves to be accepted.
 We recommend the authors to incorporate the comments and clarifications that arose during the discussions.